# Data-Driven Predictive Control of Exoskeleton for Hand Rehabilitation with Subspace Identification

**DOI:** 10.3390/s22197645

**Published:** 2022-10-09

**Authors:** Erkan Kaplanoglu, Gazi Akgun

**Affiliations:** 1Department of Engineering Management & Technology, University of Tennessee at Chattanooga, Chattanooga, TN 37403, USA; 2Department of Mechatronics Engineering, Marmara University, Istanbul 34744, Turkey

**Keywords:** DDPC, hand rehabilitation, subspace identification

## Abstract

This study proposed a control method, a data-driven predictive control (DDPC), for the hand exoskeleton used for active, passive, and resistive rehabilitation. DDPC is a model-free approach based on past system data. One of the strengths of DDPC is that constraints of states can be added to the controller while performing the controller design. These features of the control algorithm eliminate an essential problem for rehabilitation robots in terms of easy customization and safe repetitive rehabilitation tasks that can be planned within certain constraints. Experiments were carried out with a designed hand rehabilitation system under repetitive and various therapy tasks. Real-time experiment results demonstrate the feasibility and efficiency of the proposed control approach to rehabilitation systems.

## 1. Introduction

To regain the limb movement ability lost because of any disease or accident, a repetitive and intense rehabilitation process is required. Physiotherapists treat patients during this process. They frequently use rehabilitation robots to help patients perform the right movements with the right intensity while under control inside or outside of the clinic [1]. In recent years, academic or R&D studies were conducted frequently for design and implementation of rehabilitation robotics. Exoskeletons, which allow for direct limb manipulation, or end-effect robots, that support therapy by manipulating the limb’s distal point, are two mechanical structures that are used in rehabilitation robot design. Specially designed hand rehabilitation robots are used to treat post-stroke hand movement limitations. These robots need to be made in accordance with the complex structure of the hand, which has about 20 degrees of freedom; it should also be supported by a robust and adaptive control algorithm so that it can function consistently for each patient [2,3,4,5]. Devices that support active rehabilitation can be used during the patient’s completely lost movement during the passive phase of the repetitive rehabilitation process. The rehabilitation robot should perform fewer movement tasks as the patient recovers more and give them more responsibility. It should assist in this instance with active-assistive rehabilitation procedures. Resistance exercises for muscle strengthening can be done once the patient regains his mobility [6,7]. Robot control design must have an adaptive structure to perform the rehabilitation processes. Active power control is not possible for the direct trajectory control robots described in the literature. Rehabilitation robots are controlled to carry out active, active-assistive, and passive rehabilitation tasks using control algorithms, such as impedance or admittance control [8,9,10].

Systems with nonlinear structured uncertainties can also be controlled using variable structure controllers. These controllers allow for the use of parametric perturbations between lower and upper limits to deal with complexity and noise from the external world [11]. The control of a lower extremity exoskeleton was accomplished in [12] using the anti-disturbance sliding mode controller. Additionally, a robust adaptive sliding mode controller is proposed in [13] to deal with unknown and bounded dynamic uncertainties of the upper limb exoskeleton for passive rehabilitation tasks. A fuzzy approximation-based backstepping control is another approach to the control of an exoskeleton for rehabilitation [14,15].

System identification is crucial for establishing the controller design. The kinematic and dynamic parameters must be accurately determined, and the robot must generate the force required by the patient for it to adapt to the kinematic structure of each patient [16,17].

The limb that the robot works on simultaneously during rehabilitation is breakable and sensitive. While manipulating this limb, the robot must avoid harming it. Both the patient and the robot must cooperate, and the robot can only move the patient’s hand within limits. In this instance, the control algorithm should find the best solution—not just any solution—for the targeted task while taking the constraints into account. Algorithms for model predictive control can be employed to achieve this [2,18].

Model-based control algorithms, such as model predictive control, are frequently used in process control applications because they produce the optimal solutions possible, given the constraints. These algorithms can also be used for rehabilitation procedures because they use past data collected over a specific horizon to determine which estimation model offers the best solution [19,20]. For every patient, a unique model will be developed with unique system parameters and dynamics. A universal model cannot be used in this case. Since MPC algorithms are model-based algorithms, continuous identification data-driven predictive control algorithms can be applied in this situation [21].

The subspace-based parameter estimation approach is used in this study to estimate system parameters. The parameters appropriate for the model defined along a horizon are obtained using the state data and control signal gathered during open loop operation along this horizon [22]. A predictive control algorithm and an optimal control rule are built using the model and parameters acquired here. The control rule is handled independently of the constraints, and experiments are carried out to determine how much each model variable contributes to the control rule’s success.

The rest of the paper is organized as follows: Section 2.1 introduces the suggested control strategy, followed by Section 2.2 and Section 2.3, which describe the subspace identification method and data-driven predictive controller, respectively. The experimental setup is also described in the Section 2. The Section 3 of the paper includes the experimental results. This section examines and presents the impact of all parameters on the success of the DDPC. Finally, every result is discussed and concluded.

## 2. Materials and Methods

### 2.1. Exoskeleton for Hand Rehabilitation

The mechanical structure of the exoskeleton that was previously proposed is effective with the simple design. In this structure, the middle and proximal phalanxes serve as links for two 4-bar mechanisms that are sequentially coupled. Additionally, the metacarpal phalangeal (MCP) joint has a range of motion of 55 degrees of flexion, whereas the proximal interphalangeal (PIP) joint has a range of motion of 65 degrees of flexion from a fully extended posture. The linear actuator may move at a maximum speed of 12 mm per second and a maximum force of 45 N. A full hand opening or closing action requires roughly 5 s for a stroke of 50 mm. The mechanical structure and biodynamic fit of the hand make the designed system more practical in terms of usage and productivity than similar ones [2,23,24].

The system consists of L0, L1, L2 linkages. The MCP joint angle ($\phi_{1}$) and PIP joint angle ($\phi_{2}$) rely on the length of the linear actuator ($l_{0}$), as shown in Figure 1. Therefore, simultaneous actuation of both joints occurs. Whenever a new user (subject or patient) puts on the exoskeleton on their hand, the exoskeleton and the finger unite to form a single, distinctive 1-DOF system.

### 2.2. Proposed Control Method

In this study, position tracking was performed with a DDPC designed using subspace identification and model predictive control (MPC) techniques. MPC is a model-based technique that was successfully applied over the years. The difficulties (cost, time, and effort) associated with the identification of a predictive model of the system are major barriers that prevent the widespread adoption of MPC for complex systems.

It is challenging to describe the system model when using rehabilitation robots, orthotics, and prosthetic applications, because the controlled system’s parameters vary depending on the patient or the healing process. Therefore, data-driven algorithms are chosen over model-based algorithms. Model-based control is often a time-consuming and expensive process. Additionally, it is ineffective for a system that will be applied to multi-user processes with various mechanical characteristics. The benefit of the DDPC method is that it contains historical data continuously, making the system sensitive to any changes in mechanical parameters. The need for memory is made clear by the necessity of storing historical data for modeling. This may also be viewed as a drawback.

The input and output signals obtained from open loop data are used as input and output variables in the DDPC. The obtained sub-space matrices can be used to calculate the DDPC algorithm weight parameters. During the specified horizon, the system can keep track of the reference using a control rule based on past input and output data as well as tracking errors. By closely monitoring all activity along the horizon, the rule based on previous data will be able to react quickly. Particularly, the fast and accurate identification of new circumstances will enable quick adaptation. The structure of the proposed control method is shown in Figure 2.

### 2.3. System Identification Methods

#### 2.3.1. Output Error Method for Identification

The purpose of the output error method is to find the best parametric model according to the given specific criteria. The criteria is the error between the measured noisy output and the simulated model output, as shown in Equation (1).
(1)$$\epsilon =x_{2}-{\hat{x}}_{2}$$

Here, ${\hat{x}}_{2}$ is the estimated state variable of the system calculated using estimated values of unknown dynamic parameters. If the parallel model configuration of the model is similar to 2, we can find the parameter with equations in 3.
(2)$${\dot{\hat{x}}}_{2}=-\hat{a}{\hat{x}}_{2}-{\hat{f}}_{s}sgn\left({\hat{x}}_{2}\right)+\hat{b}u,\;{\hat{x}}_{2}\left(0\right)={\left({\hat{x}}_{2}\right)}_{0}$$
where *a*, *b*, and $f_{s}$ are the unknown parameters to be identified. In this equation, $x_{2}$ is the state variable, and *u* is the input of the system, and both are measured and/or observed.
(3)$$\dot{\hat{a}}=-\gamma_{1}\epsilon x_{2},\;\dot{\hat{b}}=\gamma_{2}\epsilon u,\;\;{\dot{\hat{f}}}_{s}=\gamma_{3}\epsilon sgn\left(x_{2}\right)\;$$
where $\gamma_{1}$, $\gamma_{2}\;\mathrm{and}\;\gamma_{3}$ are the gain of the error effected to parameter. The parameters *a*, *b,* and $f_{s}$, estimated by the output error method, were studied and compared in [2] previous studies.

#### 2.3.2. Subspace Identification Method

Background information on the subspace identification matrices from open-loop data is provided in this subsection. The following section will use these matrices to design a data-driven predictive controller. We will begin by defining the system’s state-space model. As a result, the equations below can be written in state-space form for a linear discrete time-invariant system:(4)xk+1=Axk+Buk+Kek
(5)yk=Cxk+Duk+ek
where uk∈ℝm, yk∈ℝl, and xk∈ℝm are the input variables, the output variables, and the state vector variables of the system, respectively; ek∈ℝl is white noise disturbance. The system matrices A∈ℝn×n, B∈ℝn×m, C∈ℝl×n, D∈ℝl×m, and K∈ℝl×l are the state, input, output, feed-through, and Kalman filter gain matrices of the system, respectively.

We assume that the measurements of the inputs uk and the outputs yk for k∈1, 2, …, N are available for identification. The input Hankel matrices for uk are represented as Up and Uf.
(6)Up≜u1u2⋯uN−2M+1u2u3⋯uN−2M+2⋮⋮⋱⋮uMuM+1⋯uN−M+1
(7)Uf≜uM+1uM+2⋯uN−M+1uM+2uM+3⋯uN−M+2⋮⋮⋱⋮u2Mu2M+1⋯uN
where the subscripts ‘*p*’ and ‘*f*’ represent the ‘past’ and ‘future’ matrices of the variables. Similarly, Hankel matrices for yk, represented as Yp and Yf defined as (8) and (9), respectively. The dimensions of the matrices are Yp, Yf∈ℝMl×N−2M+1 and Up, Uf∈ℝMm×N−2M+1, respectively.
(8)Yp≜y1y2⋯yN−2M+1yy3⋯yN−2M+2⋮⋮⋱⋮yMyM+1⋯yN−M+1
(9)Yf≜yM+1yM+2⋯yN−M+1yM+2yM+3⋯yN−M+2⋮⋮⋱⋮y2My2M+1⋯yN 

These data block Hankel matrices are made rectangular in the subspace identification method to reduce the undesirable effects of noise on the identification system. This situation can be achieved via having a large set of data, denoted by the variable N. Moreover, M in Equations (6)–(9) can be understood as the order of the predictor equation. For a successful identification of the system behavior, the order M must be bigger or at least equal to the real system order *n* as manifested in the dimension of the state matrix A [25]. The system’s past and future state vectors are written as:(10)$$X_{p}\;≜\;\left[x_{1}\;x_{2}\;\cdots \;x_{N-2M+1}\right]$$
(11)$$X_{f}\;≜\;\left[x_{M+1}\;x_{M+2}\;\cdots \;x_{N-M+1}\right]$$

As a result of the derivation of Equations (4) and (5), the equations can be written as below. These equations are known as the subspace matrix input–output equations used for identification [26,27].
(12)$$Y_{f}=\Gamma_{M}X_{f}+H_{M}^{d}U_{f}+H_{N}^{s}E_{f}$$
(13)$$Y_{p}=\Gamma_{M}X_{p}+H_{M}^{d}U_{p}+H_{N}^{s}E_{p}$$
(14)$$X_{f}=A^{M}X_{p}+\Delta_{M}^{d}U_{p}+\Delta_{M}^{s}E_{p}$$

ΓM∈ℝMl×n can be described as the extended observability matrix, ΔMd∈ℝn×Mm as reversed extended controllability matrix (deterministic), and ΔMs ∈ℝn×Ml as the reversed extended controllability matrix (stochastic) [20,28]. Yf can be written with Equations (12)–(14) as below:(15)Yf=ΓmAmΓm†−AmHm+ΔmYpUp+HmUf+(ΔMs−AmΓm†HMs)Ep+HMsEf

Since the effect of Ef is constant white noise, and cause of the stability of a Kalman filter, Equation (15) can be written to give an optimal prediction expression of the system output Yf as follows:(16)Y^f=LwWp+LuUf
where Wp=YpUpT, Uf consists of past inputs and outputs and future inputs, respectively. Lw∈ℝMl×Ml+m is the subspace matrix corresponding to the past input and output states, and Lu∈ℝMl×Mm is the subspace matrix corresponding to the future inputs. Future output values in Equation (16), as well as the system’s future input, can be formulated as a linear combination of the system’s past input and output states. The system’s behavior will then be described using Equation (16), rather than going back to the identification techniques that yield the traditional transfer function or state-space description of the system.

The following least squares problem is solved to calculate $L_{w}$ and $L_{u}$ from the Hankel matrices.
(17)$$min||Y_{f}-\left[\begin{array}{cc} L_{w} & L_{u} \end{array}\right]{\left[\begin{array}{c} W_{p}\\ U_{f} \end{array}\right]||}_{F}^{2}$$

This problem can be solved from the orthogonal projection of the row space of $Y_{f}$ into the row space of the matrices $W_{p}={\left[\begin{array}{cc} Y_{p} & U_{p} \end{array}\right]}^{T}$. This can be defined by Equation (18) as follows:(18)$${\hat{Y}}_{f}=Y_{f}/\left[\begin{array}{c} W_{p}\\ U_{f} \end{array}\right]$$

### 2.4. Data-Driven Predictive Controller

The data-driven predictive control algorithm uses the linear subspace predictor and the cost function of MPC algorithm.

M and N are lengths of data. Furthermore, $y_{d}\left(k+1\right)$, $y\left(k+1\right)$, $u\left(k-M\right)$ are the desired output r, output, and the input, respectively. All I/O data are stored in a database and then can be used again in control.

The MPC algorithm cost function form [29,30] can be written with the prediction and control horizon $N_{p}$ and $N_{c}$ equal to f as follow:(19)$$J=\left(\overset{N_{p}}{\underset{k_{p}=1}{\sum }}{\left({\hat{Y}}_{t+k_{p}}-r_{t+k_{p}}\right)}^{T}W_{Q}\left({\hat{Y}}_{t+k_{p}}-r_{t+k_{p}}\right)+\overset{N_{c}}{\underset{k_{c}=1}{\sum }}\Delta U_{t+k_{c}}^{T}W_{R}\Delta U_{t+k}\right)$$
where $W_{Q}$ and $W_{R}$ are the weight matrices, $r_{t}$ is the reference signal at the current time t, $N_{p}$ and $N_{c}$ are the prediction and control horizon, respectively. $N_{c}$ maybe less than or equal to the prediction horizon $N_{p}\;\left(N_{c}\leq N_{p}\;or\;N_{c}\leq f\right)$.

We maintain to improve the basics of DDPC via rewriting the cost function of MPC Equation (19) in quadratic form. Using Equation (16) and the reference signal of $r_{t}$, we can update the cost function as follows:(20)$$J={\left(L_{w}\Delta W_{p}+L_{u}^{N_{c}}\Delta U_{N_{c}}+Y_{t}-r_{t+1}\right)}^{T}W_{Q}\times \left(L_{w}\Delta W_{p}+L_{u}^{N_{c}}\Delta U_{N_{c}}+Y_{t}-r_{t+1}\right)+\Delta U_{N_{c}}^{T}W_{R}\;\Delta U_{N_{c}}$$

If we solve the cost function, the control rule can be written as follows:(21)$$\Delta U_{N_{c}}=-{\left({\left(L_{u}^{N_{c}}\right)}^{T}W_{Q}\left(L_{u}^{N_{c}}\right)+W_{R}\right)}^{-1}\times {\left(L_{u}^{N_{c}}\right)}^{T}W_{Q}\left(L_{w}\Delta W_{p}+\left(Y_{t}-r_{t+1}\right)\right)\;=-K_{\Delta W_{p},N_{c}}\Delta W_{p}-K_{e,N_{c}}\left(Y_{t}-r_{t+1}\right).$$
where $-K_{\Delta W_{p},N_{c}}$ and $K_{e,N_{c}}$ are the weight of the past data vector and the weight of the tracking error, respectively.

At each time condition, only the first element of $\Delta U_{N_{c}}$ is used to calculate the control input $U_{t+1}$, which complies with $\Delta U_{t+1}$. Hence, abbreviating the first m rows in Equation (21) gives as below:(22)$$\Delta U_{t+1}=-K_{\Delta W_{p}}\Delta W_{p}-K_{e}\left(Y_{t}-r_{t+1}\right)$$

With,
(23)$$K_{\Delta W_{p}}=[I_{m}\;0_{m\times \left(M-1\right)m}]K_{\Delta W_{p},N_{c}}$$
(24)$$K_{e}=[I_{m}\;0_{m\times \left(M-1\right)m}]K_{e,N_{c}}$$
where $I_{m}$ is an identity matrix of size m while $0_{i\times j}$ is a zero matrix with i rows and j columns. Consequently, the control input $U_{t}$ can be written as follows:(25)$$U_{t}=U_{t-1}+\Delta U_{t}$$

### 2.5. DDPC with Considering Constraints via Quadratic Programming

The ability of MPC and other predictive control algorithms to include constraints in the final control solution is one of their advantages. In this section, a constrained DDPC algorithm is provided, taking the constraints in Equation (26) into account.
(26)$$F_{m}\Delta u_{min}\leq \Delta u_{Nc}\leq F_{m}\Delta u_{max}\;F_{m}u_{min}\leq u_{Nc}\leq F_{m}u_{max}\;F_{l}y_{min}\leq \Delta y_{f}\leq F_{l}\Delta y_{max}\;F_{l}y_{min}\leq y_{f}\leq F_{l}y_{max}$$

Here Fm and Fl are defined as Fm=ImIm…ImT∈ℝNcm x m, Fl=IlIl…IlT with identity matrices Im and Il. Further, uNc=ut+1Tut+2T…ut+NcTT.

The control signal is optimized in the constrained DDPC algorithm while taking into account the constraints placed on the cost function specified in the earlier sections. The inequalities shown in Equation (27) are reached when the constraints defined in Equation (26) are rewritten as a function of [2,20].


(27)
$$\underset{A_{QP}}{\underset{︸}{\left[\begin{array}{c} \begin{array}{c} I_{N_{c}m}\\ -I_{N_{c}m}\\ \Gamma_{m} \end{array}\\ \begin{array}{c} -\Gamma_{m}\\ L_{u}^{Nc}\\ -L_{u}^{Nc} \end{array}\\ \begin{array}{c} \Gamma_{l}L_{u}^{Nc}\\ -\Gamma_{l}L_{u}^{Nc} \end{array} \end{array}\right]}}\Delta u_{Nc}\leq \;\underset{b_{QP}}{\underset{︸}{\left[\begin{array}{c} \begin{array}{c} F_{m}\Delta_{u_{max}}\\ -F_{m}\Delta u_{min}\\ F_{m}u_{max}-F_{m}u_{t} \end{array}\\ \begin{array}{c} -F_{m}u_{min}+F_{m}u_{t}\\ F_{l}\Delta y_{max}-L_{w}\Delta w_{p}\\ -F_{l}\Delta y_{min}+L_{w}\Delta w_{p} \end{array}\\ \begin{array}{c} F_{l}y_{max}-F_{l}y_{t}-\Gamma_{l}L_{w}\Delta w_{p}\\ -F_{l}y_{min}+F_{l}y_{t}+\Gamma_{l}L_{w}\Delta w_{p} \end{array} \end{array}\right]}}$$


It can be discovered by optimizing the cost function for the DDPC algorithm given in Equation (28) with constraints.
(28)$$\underset{\Delta u_{Nc}}{\min}\frac{1}{2}\Delta u_{Nc}^{T}H\Delta u_{f}+\Delta u_{Nc}^{T}f\;s.t.\;A_{QP}\Delta u_{Nc}\leq b_{QP}$$

The quadratic programming (QP) algorithm can be used to solve this optimization process. The QP algorithm determines the ideal control signal $\Delta u_{Nc}$ while accounting for the constraints.

### 2.6. Experimental Setup

For the execution of the parameter estimation algorithms, the Matlab/Simulink environment is used. STM32F4107 CPU was used to run control algorithms. The dual-channel H bridge L298 driver IC was used for DC motors and a 9V Li-Po battery was used as the voltage supply for the motor. The Matlab/Simulink environment was also utilized during the studies to gather and store data. Communication is established between the microprocessor and the Matlab/Simulink environment with the UART protocol. In each experiment, control signals or reference position values were sent from the Matlab/Simulink environment to the CPU using the serial communication protocol. The employed experimental setup is shown in Figure 3.

The data-driven predictive control method was validated on a real-time system by different experiments. The system position was directly measured using a linear potentiometer.

### 2.7. Passive and Active Rehabilitation

The design of both active and passive rehabilitation tasks requires estimation or measurement of the external force exerting on the exoskeleton’s endpoint. In this study, a micro load cell is placed between end point of the linear actuator shaft and fork joint to measure external force ($F_{ex})$. External force can be used to stimulate a virtual mass-spring-damper system, as shown in Figure 4 and its mechanical parameters can be adjusted depending on the type and degree of rehabilitation required.

When the measured external force is applied to the virtual system using Equation (29), the virtual system’s response $x_{d}$, can be found.
(29)$$m_{d}{\ddot{x}}_{d}+b_{d}{\dot{x}}_{d}+k_{d}x_{d}=F_{ex}$$

For active rehabilitation task, the virtual system’s position response, $x_{d}$, can be used to deviate to the desired reference from the actuator’s actual position, $x_{1}$ (Equation (30)). In this instance, the behavior of the controller is a regulator and keeps the actuator in its actual stroke position. If the external force is greater than zero, the desired reference is different from the actual position of the actuator.
(30)$$x_{r}^{\prime }=x_{1}-x_{d}$$

For passive rehabilitation, $x_{d}$ can be used to create a deviation from the predefined trajectory $x_{r}$, as shown in Equation (31), and the controller can make use of this desired reference.
(31)$$x_{r}^{\prime }=x_{r}-x_{d}$$

It is possible to decide how the virtual mass-spring-damper system responds to external force and carries out passive, active, or assistive rehabilitation tasks by setting the parameters within the acceptable range of values.

## 3. Results and Discussion

### 3.1. Experimental Results of Subspace Prediction Algorithm

During the tests, the robot was not subjected to any external force and the exoskeleton is tested on a healthy human hand. The predicted model was compared with the state-space model obtained by the output error method; those applications and results are explained in past studies [23]. By using the model horizon parameters p=30, f=10, the subspace estimation model parameters ($Lu_{r}$ and $Lw_{r})$ are calculated. The Lu and Lw obtained with the same values as the previous p and f from the system are then calculated using the $u_{t}$ input signal to be tested, the Lu and Lw are then calculated using the same values as the previous p and f from the system (Figure 5. In the graphics, the results calculated using the reference ($Lu_{r}$ and $Lw_{r})$ models and the actual Lu and Lw models are compared.

The output error-based model (OEbM) output is compared with the sub-space model (SPbM) estimation result obtained with the *u* input signal ($u=A_{0}sin\omega_{0}t,\;\;\omega_{0}=0.5\;\mathrm{rd}/\mathrm{sn}$ ve $A_{0}=5$) shown in the Figure 6. The linear actuator’s stroke length, which is considered the system output, is represented by the *y*-axis on the graph as position.

The percentage error function in Equation (32) was obtained between the OEbM output, which was used as the reference model, and the SPbM output to evaluate the outcomes of the model estimate test, and this number was chosen as the performance criterion.
(32)e=∑i=1nyOEbM−ySPbMyOEbMn

This experiment’s error value (e) was calculated as 1.1871%. Table 1 provides the percentage error values of the test results with additional specified input signals.

The range of $\omega_{0}=0.3\;\mathrm{rd}/s$, where the input signal runs throughout the full stroke in a single alternate, and $\omega_{0}=2\;\mathrm{rad}/s$, which gives 5% displacement on the motor stroke, is tested using a single frequency component. The first eight experiments revealed a linear correlation between the modeling success and the parameters of the frequency component of the input signal. As a result, success increased and error decreased at higher frequency values, while success increased at lower frequency values. Since the full stroke length could not be studied in the experiments with high frequency *u* input signals, it can be argued that all the system’s characteristics could not be apprehended in them. This has an impact on modeling success.

The following experiments are performed with a sinusoidal input signal with two separate frequency components ($u=A_{0}sin\omega_{0}t+A_{1}sin\omega_{1}t)$. The experiment that was performed using input signals with $\omega_{0}=0.2\;\mathrm{rd}/s$ and $A_{0}=2$, $\omega_{1}=0.8\;\mathrm{rd}/s$ and $A_{1}=3$ (Experiment 10) results are shown in Figure 7.

It was noted that in experiments using the *u* input signal with two frequency components, the average success rate increased. When experiments 9 through 12 in Table 2 are examined, it becomes clear that the error function produces results that are similar across the ranges of experimental parameters. It was noted that the error function is negatively impacted by the separation between the two component frequencies.

The following experiments (13–16) use *u* input signals that have three distinct frequency components ($u=A_{0}sin\omega_{0}t+A_{1}sin\omega_{1}t+A_{2}sin\omega_{2}t),$ as shown in Figure 8. The experiments conducted are more useful than the earlier experiments, as shown in Table 3. Different frequency components are found to increase the success of modeling.

The following tests were conducted using input signals that contained a scanning frequency as shown in Figure 9. These input signals are configurations for input signals that begin with low-frequency components and increase throughout the designated range. The average success is higher in these experiments, as shown in Table 4. This input signal, which has components in several different frequency ranges, helps to clarify the system’s characteristics.

### 3.2. Experimental Results of Data-Driven Predictive Controller

The predicted model parameters and control parameters $Ke\;\mathrm{and}\;K\Delta wp^{1x2p}$ are calculated. The step function and a sinusoidal trajectory response (2 *π* radians from position 0 to 50%) are examined to analyze the parameters affecting the control algorithm, and the results are discussed. 

#### 3.2.1. Effect of Data Length on Control Success

Figure 10 shows the relationship between the calculated control coefficients and the control success as a function of the data size of the input signal (*u*), which is used as the estimation input signal. In experiments, only the input signal *u*’s data length is changed, while the other parameters, p=30,f=10,Q=1,R=2, and Nc=5, remain constant. As a result, it was found that as the number of data increases, overshoot decreases, increasing the success of the control. The control coefficients from experiments with the same frequency components but fewer data are seen to negatively affect the success as the number of data decreases. The overshoot increases as *N* decreases. The quantity of data has no meaningful effect on the rise time.

#### 3.2.2. Analysis of Model Horizon Parameters (*p*, *f*)

The system state equations along the chosen horizon are the basis of the subspace-based estimation model. Using the input signal from the most successful experiment (Experiment 20) from the previous test results, the test results referring to the horizon parameter success characteristic are presented in this section.

As a result of parameter estimation using various historical model horizon values, the control parameters are displayed in Table 5. Experiments are carried out with constant parameters f=10,Q=1,R=2, and Nc=5 by varying the model horizon *p* between 30 and 60. The effect of the previous model horizon on the system response is investigated in these experiments. Rise time and percent overshoot are used as performance factors in step function experiments. The success factor in trajectory tracking is the mean of the squares of the errors. Table 5 details all these values.

Without considering the experiment taken as *p* = 50 in the table, it can be said that the other values show that the step response and trajectory tracking success are negatively impacted by an increase in the *p* value as shown in Figure 11. However, in the experiment where *p* is set at 50, success increases once more. The control is seen to be totally broken in the following experiment. This assumes that there might be a linear relationship between control success and the past modeling horizon, *p*. When the response of the system to the sine waveform is analyzed, it is seen that even the worst result (*p* = 60) follows the trajectory with a certain error (mse = 16.96 from Table 5).

In the following experiments, the past model horizon *p,* and other parameters (p=30,Q=1,R=2, and Nc=5) are held constant to examine the control success of the future model horizon *f*. Rise time and percent overshoot are considered performance factors in experiments using the step function. The average of the squares of the errors is used as a performance metric for trajectory tracking success. In Table 6, each of these values is described in detail.

When the tables and graphics are examined, it is observed that there is a nonlinear relationship between the model future horizon *f* and control success. The overshoot and rise times are close to each other in experiments where the *f* value of the future horizon is between 20 and 25. The increase in the mean value and standard deviation of *Ke* and KΔwp has a negative effect on the rise time. The rise time increases as the *f* parameter increases, but the average overshoot decreases. The response of trajectory also shows that the overshoot is high in the experiment where *f* is set to 5 (Figure 12). This overshoot is a result of the *Ke* value obtained in this experiment being much lower than in the other experiments. Based on gathered data, it is seen that, even if the number of future horizons is chosen as only 5, the steady-state stability is considered as critically stable.

The tables display control coefficients based on the parameters used in the experiments. The graph in Figure 13 shows the control parameters (*Ke*) for all *p* and *f* values in the determined range (5–100). When the *Ke* value is between 0.2 and 0.3, it is apparent from the experiments, the results of which are evaluated, that the success rate is high. Additionally, the tables display the mean and standard deviation values of the KΔwp parameter calculated in successful results. Figure 13 and Figure 14 can be used to analyze the suitable numeric selections of *p* and *f* parameters for these values. The color scale of the *p* and *f* pairs providing successful *Ke* values is shown in Figure 13. It is clear from all the graphs that the values of *p* and *f* for the successful control task can be selected from a range of 5 to 80 and 20 to 60, respectively.

### 3.3. The Effect of Q and R Parameters on the Control System Succession

It is shown in equation 16 that the parameters *Q* and *R* in the general MPC cost function are the weights of the reference error and the differential control signal, respectively. These weights have a direct impact on the success of the control because the data-driven predictive control rule is based on the MPC cost function. The success of trajectory tracking and step response are examined using the experiments in this section with the *Q* and *R* parameters. With *p* = 30, *f =* 10, and *Nc* = 5, various *Q* and *R* parameters are tested in experiments.

The *Q* parameter and the rise time have a linear relationship, as can be seen when Table 7 is examined. In addition, the ratio of the *Q* parameter to *R* also affects the rise time. It is clearly seen that the *R* parameter affects the overshoot as seen in Figure 15. The overshoot increased up to a maximum value of 8% in the experiment where *R* was set to be 50. These tests led to the conclusion that selecting a small *Q* value would have a positive effect on the rise time as shown in Figure 16. The experiments also confirmed that selecting a low value (*Q* = 1) and choosing the parameter *R* within the optimal ranges (10,200) have a positive impact on the response of step and trajectory tracking.

The control horizon *Nc* in the data-driven predictive control algorithm is described in Equation (16). Unlike *f*, which is the model horizon, *Nc* specifies the size of the space from which the system control signal $u_{c}$ should be calculated. It is used by the control algorithm to compute the system control signal $u_{Nc}$, which is the first component of $\Delta u_{Nc}$ along *Nc*. These tests were conducted with the following parameters: p=30, f=10, Q=1, and R=2. The impact of *Nc* on step function response and trajectory tracking response is shown in Table 8 and Figure 17. It is clear from the tables and graphs that the rise time has a linear relationship with *Nc*’s trajectory tracking response and step function response.

### 3.4. Passive and Active Rehabilitation

The patient is completely passive during passive rehabilitation tasks. The patient’s hand is completely guided by the exoskeleton. The patient did not exert any force on the exoskeleton for the first 10 s, as shown in Figure 18. The controller in this case uses $x_{r}^{\prime }=x_{d}$. When a maximum counterforce of 7–8 N was applied in the final 10th second, the virtual system response $x_{d}$ deviates from the reference $x_{r}^{\prime }$. This makes it possible for the system to react to the patient in accordance with any unexpected issues that might arise on the patient’s finger. The response stiffness of the system to the patient’s hand can be adjusted by appropriately adjusting the parameters of the virtual mechanical system as shown in Equation (26).

The controller functions as a regulator during the active rehabilitation process and tries to keep the current position of the actuator stroke. The reference shifted, as shown in Figure 19, because of the patient applying an external force of about –6N in the 12th second. The patient made the flexion movement with a stable force applied to the system. The extension movement was carried out by exerting force in the opposite direction after the 14th second. The system can be operated at greater forces by modifying the virtual mechanical system’s parameters. The system can now perform in resistive mode as a result. The system operates in assistive mode by taking $x_{d}$ in negative.

The study of the robustness of MPC can be approached in a variety of ways. The first focuses on the closed-loop systems’ robustness when created utilizing the nominal system (i.e., neglecting uncertainty). The second makes an effort to achieve robustness within the context of conventional model predictive control by taking into account all feasible realizations of the uncertainty. The third approach addresses this by introducing feedback in the min–max optimal control problem solved online [31]. In this study, according to the results obtained in some experiments, it was evaluated that the system is robust against uncertainties. For example, in the analysis of model horizon parameter experiments, it was observed that the system followed the trajectory even at the worst coefficients (*p* = 60, *f* = 5). The model control values, *Ke* and *KΔwp*, are determined using this model and calculated as optimum values, showing that the controller is robust enough to handle uncertainties.

The subspace prediction estimation results to be made with the ongoing past data online will guarantee that the controller operates feasibly and continuously during the rehabilitation tasks. Along with model parameters calculated throughout a specific horizon with sub-space prediction, the model includes all model uncertainties associated with the patient’s exoskeleton and measurement noises related to force and position. The use of data-driven estimation methods and model-free control algorithms will improve the effectiveness of studies with patients who can be evaluated in a wide range of spectrums as opposed to computing the model of a biomechanical structure, such as the human hand, with conventional approximate methods.

## 4. Conclusions

In this work, a data-driven predictive control is developed for a hand exoskeleton robot used for rehabilitation. The designed control rule was used in a set of experiments, and the results were presented. The experiments are intended to examine how the parameters affecting the suggested control algorithm influence the success of the control. A data-driven predictive control algorithm is optimization-based and certain constraints are added to the problem during the optimization process; it then suggests the best solution within the boundaries set by those constraints. The rehabilitation process aims to regain the patient’s lost mobility by having them perform repeated exercises that are suited to their situation.

In our experiments to evaluate the performance of the proposed control algorithm, data length is investigated for subspace prediction, and it is expressed that, within a certain range, data length was linearly related to modeling success. On the success of controlling for the DDPC rule, the effects of the *p*, *f*, *Nc*, *Q*, and *R* parameters are examined and discussed separately. When all the test results are evaluated, we can conclude that the suggested solution is suitable for rehabilitation processes because it offers the best solutions, while still considering some limitations.

It was shown that the exoskeleton controller operates in passive, active, and assistive modes with the benefit of an auxiliary reference created using the measured external force.

## Figures and Tables

**Figure 1 sensors-22-07645-f001:**
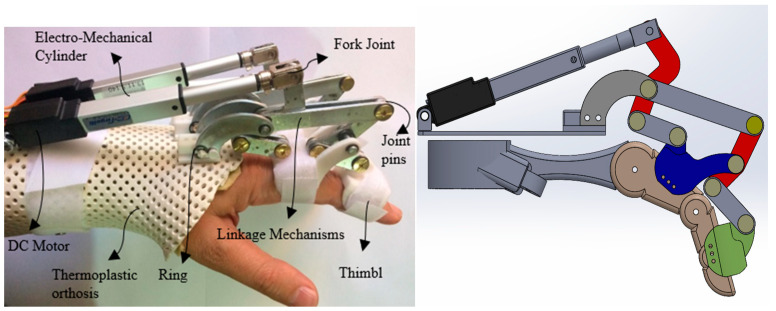
Designed exoskeleton for hand rehabilitation.

**Figure 2 sensors-22-07645-f002:**
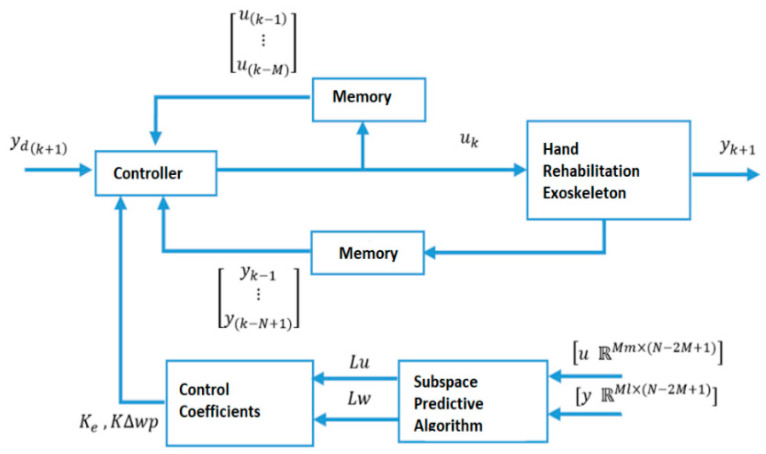
Structure of proposed data-driven predictive controller block diagram.

**Figure 3 sensors-22-07645-f003:**
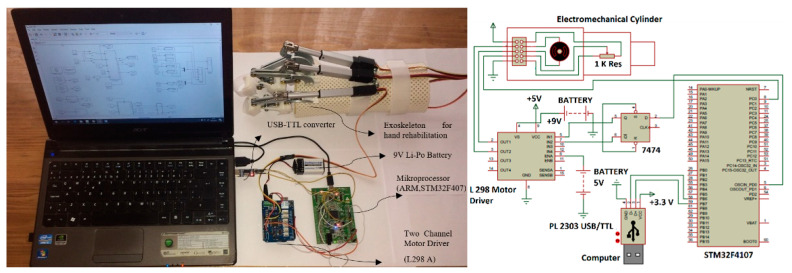
Experimental setup.

**Figure 4 sensors-22-07645-f004:**
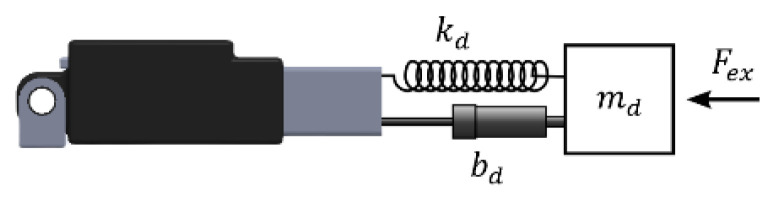
Virtual mass-spring-damper system.

**Figure 5 sensors-22-07645-f005:**
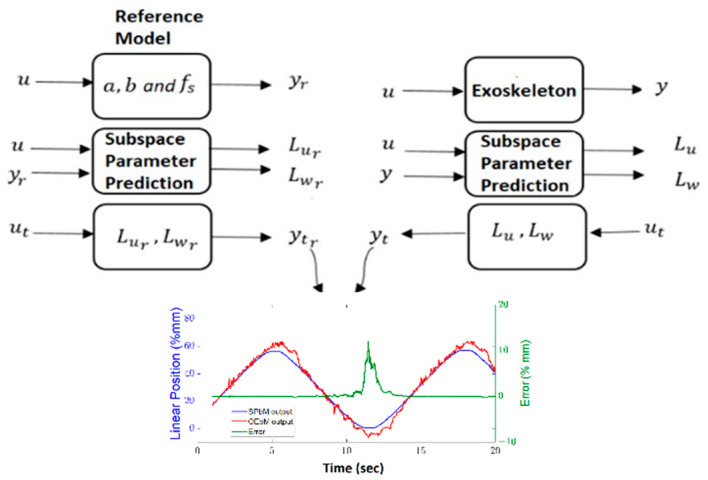
Subspace prediction validation procedure.

**Figure 6 sensors-22-07645-f006:**
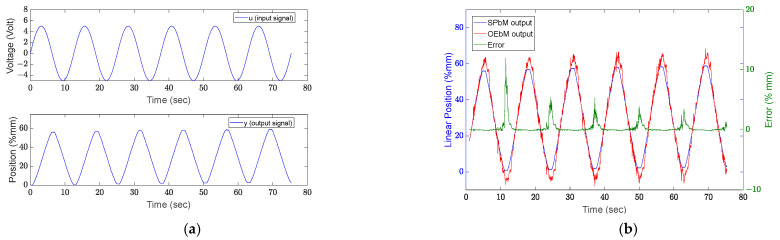
Selection of the sinus input signal $\omega_{0}=0.5\;\mathrm{rd}/s$ and $A_{0}=5$, and comparison of results. (**a**) Single sinus signal, (**b**) Comparison of results.

**Figure 7 sensors-22-07645-f007:**
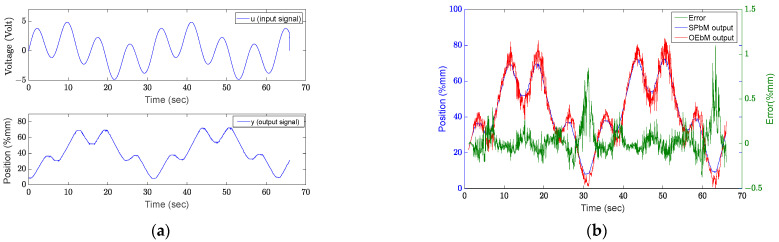
Results of experiment 10: (**a**) input and output signal, (**b**) results.

**Figure 8 sensors-22-07645-f008:**
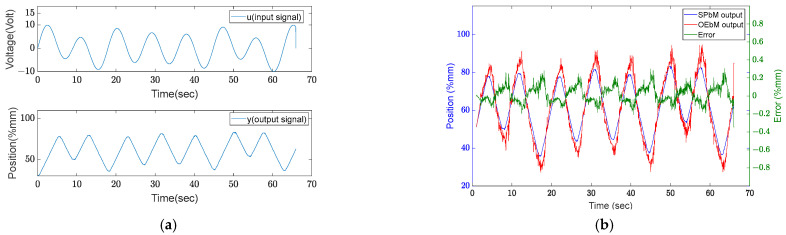
Results of experiment 10: (**a**) input and output signals, (**b**) results.

**Figure 9 sensors-22-07645-f009:**
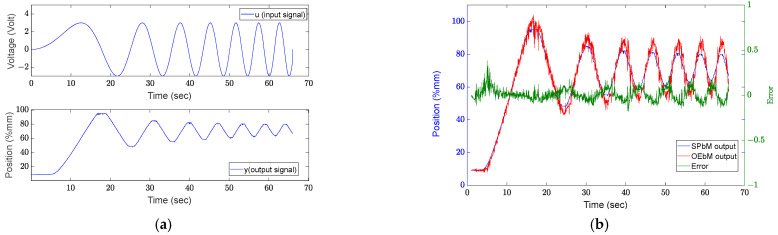
Results of experiment 20: (**a**) input and output signal, (**b**) results.

**Figure 10 sensors-22-07645-f010:**
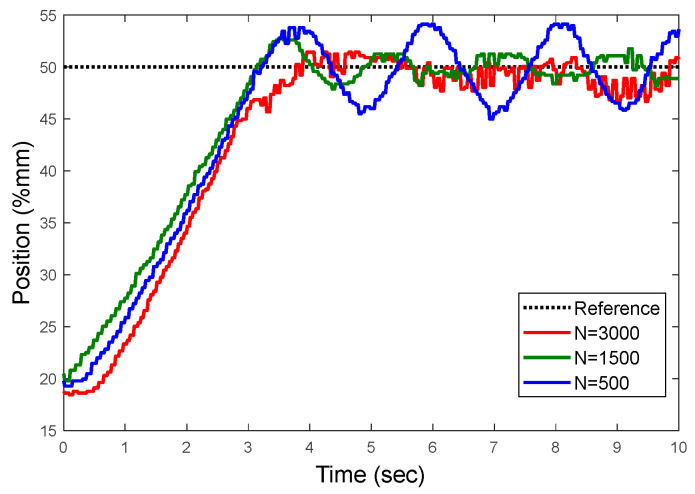
Step response according to u signals of different *N* data lengths.

**Figure 11 sensors-22-07645-f011:**
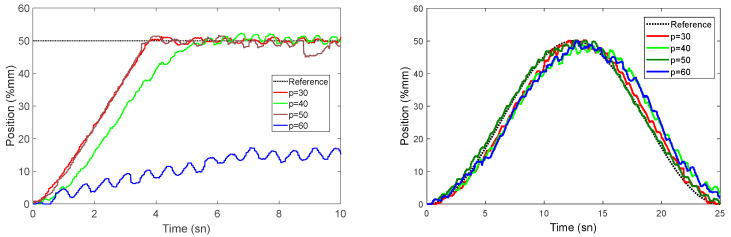
Success of different past model horizon (*p*) values in tracking trajectories and success of response of step function.

**Figure 12 sensors-22-07645-f012:**
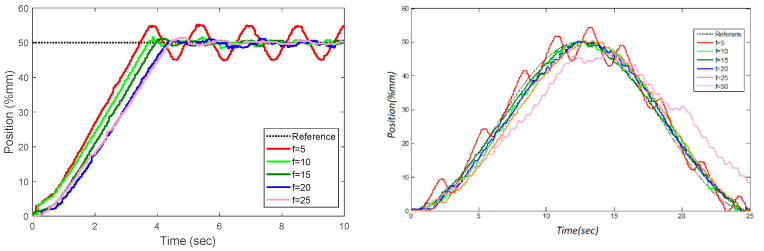
Effect of different future model horizon (*f*) values on the success of tracking the step response and trajectory.

**Figure 13 sensors-22-07645-f013:**
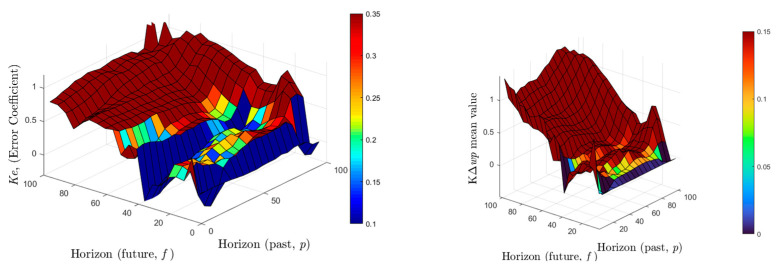
*Ke* and KΔwp calculated for the 5–100 range of *p* and *f* values.

**Figure 14 sensors-22-07645-f014:**
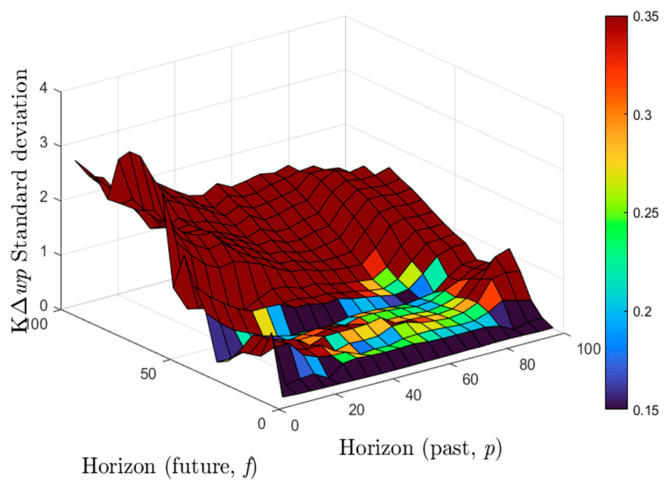
The standard deviation value of KΔwp calculated for the 5–100 range of *p* and *f* values.

**Figure 15 sensors-22-07645-f015:**
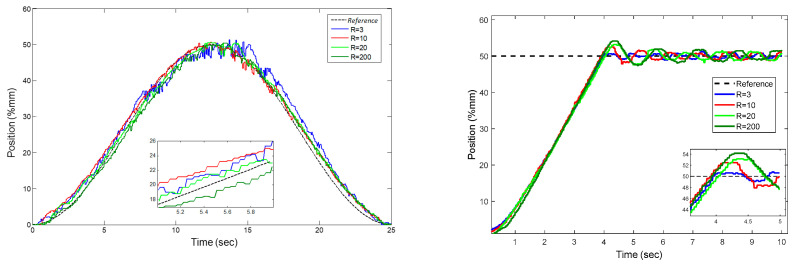
Effect of the *R* parameter on step and trajectory response.

**Figure 16 sensors-22-07645-f016:**
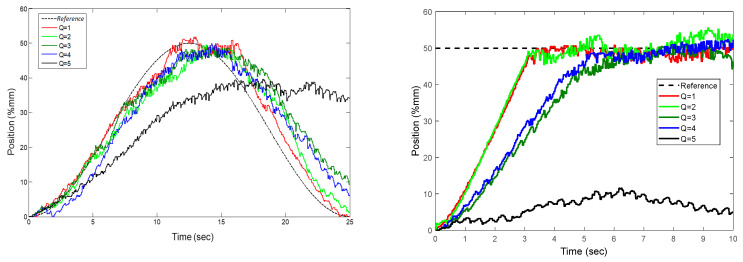
Effect of the *Q* parameter on step and trajectory response.

**Figure 17 sensors-22-07645-f017:**
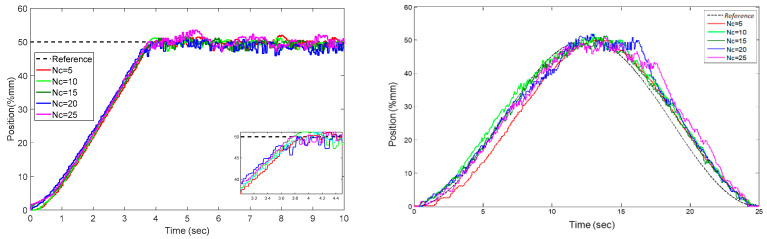
According to *Nc*, the response of step function and trajectory.

**Figure 18 sensors-22-07645-f018:**
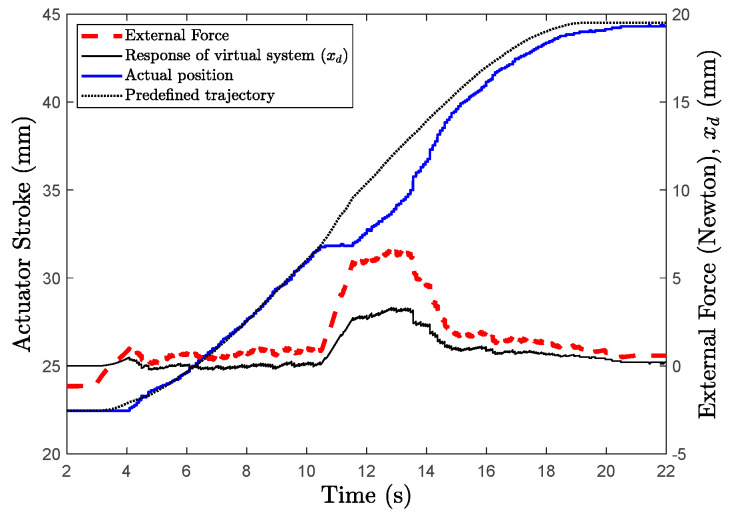
Passive rehabilitation task.

**Figure 19 sensors-22-07645-f019:**
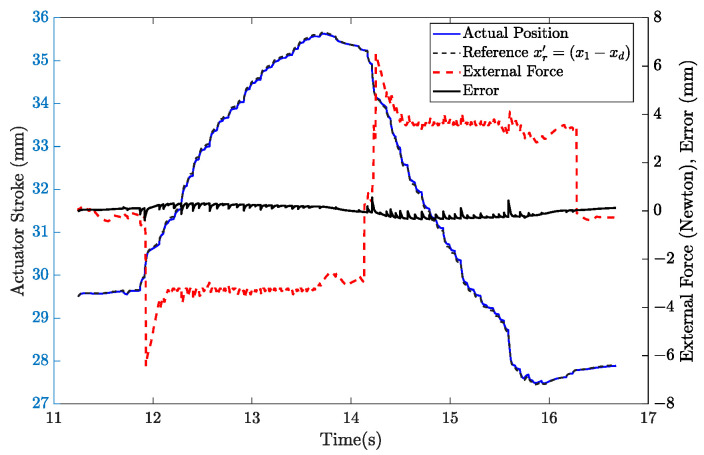
Active rehabilitation task.

**Table 1 sensors-22-07645-t001:** Experiments with a single frequency input signal $\left(u=A_{0}sin\omega_{0}t\right)$.

Experiments	$\omega_{0}\;\left(\mathrm{rd}/s\right)$	$A_{0}$	Error (e)
1	0.3	5	0.0187
2	0.4	5	0.0266
3	0.5	5	0.0337
4	0.6	5	0.0395
5	0.7	5	0.0727
6	0.8	5	0.0825
7	1	5	0.1431
8	2	5	0.9273

**Table 2 sensors-22-07645-t002:** Experiments with a two-frequency *u* input signal ($u=A_{0}sin\omega_{0}t+A_{1}sin\omega_{1}t)$.

**Experiment**	$\omega_{0\;}\left(\mathrm{rd}/s\right)$	$\omega_{1}\;\left(\mathrm{rd}/s\right)$	$A_{0}$	$A_{1}$	Error (e)
9	0.3	0.4	2	3	0.0127
10	0.2	0.8	2	3	0.0267
11	0.2	0.8	1	4	0.0360
12	0.5	0.1	2	2	0.0235

**Table 3 sensors-22-07645-t003:** Experiments with a three-frequency u input signal ($u=A_{0}sin\omega_{0}t+A_{1}sin\omega_{1}t+A_{2}sin\omega_{2}t)$.

**Experiment**	$\omega_{0}\;\left(\mathrm{rd}/s\right)$	$\omega_{1}\;\left(\mathrm{rd}/s\right)$	$\omega_{2}\;\left(\mathrm{rd}/s\right)\;$	$A_{0}$	$A_{1}$	$A_{2}$	Error
13	0.3	0.4	0.7	2	2	2	0.0088
14	0.3	0.4	0.1	2	2	2	0.0239
15	0.3	0.4	0.1	2	2	5	0.0251
16	0.5	0.4	0.1	2	2	5	0.0211

**Table 4 sensors-22-07645-t004:** Experiments with input signals that configured by scanning frequency.

Experiment	$k_{0}$	$A_{0}$	e (Error)
17	0.003	4	0.0149
18	0.005	5	0.0200
19	0.01	5	0.0083
20	0.01	3	0.0048

**Table 5 sensors-22-07645-t005:** Control parameters calculated using *p*.

Model Horizon (*p*)	*Ke*	$\mathrm{Mean}\;\mathrm{of}\;K\Delta wp^{1x2p}$	$\mathrm{Standard}\;\mathrm{Deviation}\;\mathrm{of}\;K\Delta wp^{1x2p}$	Response of Step Function		Tracking Performance
				Rising Time (s)	Overshoot %	mse
30	0.1908	0.0730	0.2499	3.88	0.17	4.3138
40	0.2853	0.0852	0.2493	5.77	0.20	18.4100
50	0.3746	0.0904	0.2399	3.84	0.18	1.4763
60	0.3780	0.1045	0.2202	* None	* None	16.9600

* In this experiment, no rise or overshoot was observed during the experiment.

**Table 6 sensors-22-07645-t006:** Control parameters defined using *f*.

Future Horizon (*f*)	*Ke*	$\mathrm{Mean}\;\mathrm{of}\;K\Delta wp^{1x2p}$	$\mathrm{Standard}\;\mathrm{Deviation}\;\mathrm{of}\;K\Delta wp^{1x2p}$	Response of Step Function		Tracking Performance
				Rising Time (s)	Overshoot %	mse
5	0.0476	0,0066	0.0222	3.38	0.0962	9.0828
10	0.1908	0.0733	0.2400	3.85	0.0322	4.3138
15	0.2191	0.0996	0.2946	3.94	0.0220	1.2027
20	0.2385	0.1580	0.3942	4.36	0.0018	3.9737
25	0.1688	0.1188	0.2980	4.48	0.0254	10.5345
50	0.1951	0.1897	0.4622	6.91	0.0524	60.1988

**Table 7 sensors-22-07645-t007:** Effect of *Q* and *R* parameters on control response.

** *Q* **	** *R* **	** *Ke* **	$\mathrm{Mean}\;\mathrm{of}\;K\Delta wp^{1x2p}$	$\mathrm{Standard}\;\mathrm{Deviation}\;\mathrm{of}\;K\Delta wp^{1x2p}$	Response of Step Function		Tracking Performance
					Rising Time (s)	Overshoot %	mse
5	2	0.7694	0.1969	0.8251	* None	* None	288.1818
4	2	0.5693	0.1513	0.6184	5.654	0.0385	43.5554
3	2	0.4975	0.1354	0.5512	6.631	0.0084	65.2003
2	2	0.4329	0.1202	0.4776	3.093	0.0688	33.0765
1	2	0.2589	0.0806	0.2981	3.031	0.0050	11.8783
1	3	0.1897	0.0648	0.2269	3.528	0.0118	6.9044
1	5	0.1293	0.0510	0.1649	3.189	0.0286	3.7065
1	10	0.0807	0.0398	0.1152	3.493	0.0490	2.7420
1	20	0.0549	0.0337	0.0889	3.902	0.0626	2.4434

* In this experiment, no rise or overshoot was observed during the experiment.

**Table 8 sensors-22-07645-t008:** Effect of *Nc* parameter on control response.

$N_{c}$	** *Ke* **	$\mathrm{Mean}\;\mathrm{of}\;K\Delta wp^{1x2p}$	$\mathrm{Standard}\;\mathrm{Deviation}\;\mathrm{of}\;K\Delta wp^{1x2p}$	Response of Step Function		Tracking Performance
				Rising Time (s)	Overshoot %	mse
5	0.1688	0.1188	0.2980	3.232	0.0254	10.5345
10	0.1799	0.0798	0.2429	3.224	0.0152	8.5312
15	0.2220	0.0768	0.2669	3.210	0.0084	3.9609
20	0.2589	0.0806	0.2981	3.180	0.0084	11.8783
25	0.3423	0.0870	0.3562	3.192	0.0684	17.3509

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
