# Peer review of "Data-Driven Predictive Control of Exoskeleton for Hand Rehabilitation with Subspace Identification"

_sensors, 2022, doi:10.3390/s22197645_

Round 1

Reviewer 1 Report

This manuscript presented the DDPC used in hand exoskeleton, and the effects of different parameters of the control algorithm were examined and discussed. It is meaningful to use the DDPC control in hand rehabilitation equipment.

There are some doubt about the content.

What is the meaning of the “position” in some of the figure which aims to show the results or success of the model? Does it means the motor stroke or others? The author should present and describe the result more accurate and detailed.

In the section of results and discussions, the author focused on the effect of different parameters and control algorithm, but it is hard to establish the relationship between the result and hand exoskeleton control. The manuscript aims to use the DDPC “for the hand exoskeleton used for active, passive, and resistive rehabilitation which was declared in the abstract. However, it is unclear about the effect of control method and parameters on exoskeleton functions, such as active, passive, and resistive rehabilitation.

There are also some minor errors,

In line 77, there are no full spelling of the abbreviations of MCP and PIP.

The labels of mechanical schematics in Figure 1 is not clear, and the description about the mechanism in line 84-88 is not sufficient.

In Figure 7(a), the labels on the horizontal and vertical axes are incorrectly spelled.

Figures 7 and 8 are not referenced in the manuscript.

In line 347, Figure 3.25 seems misspelled.

Author Response

Comment 1 from reviewer 1:  “What is the meaning of the “position” in some of the figure which aims to show the results or success of the model? Does it means the motor stroke or others? The author should present and describe the result more accurate and detailed.”

Respond 1 to reviewer 1:

The linear actuator's stroke is represented by the “position” term on the y-axis in the figures. The following has been added to line 290 to describe a definition that was missing: "The stroke length of the linear actuator, which is considered the system output, is represented by the y-axis on the graph as position."

Comment 2 from reviewer 1:  In the section of results and discussions, the author focused on the effect of different parameters and control algorithm, but it is hard to establish the relationship between the result and hand exoskeleton control. The manuscript aims to use the DDPC “for the hand exoskeleton used for active, passive, and resistive rehabilitation which was declared in the abstract. However, it is unclear about the effect of control method and parameters on exoskeleton functions, such as active, passive, and resistive rehabilitation.

Respond 2 to reviewer 1:

I am grateful for the criticism because it allowed me to refine the research. To evaluate how well the suggested methods performed during the rehabilitation process, the material and method section has been expanded to include a section on passive and active rehabilitation as subsection 2.7. In this section, the rehabilitation strategy is explained. The Results and Discussion section is expanded to include the 3.4 Passive and Active Rehabilitation section, and the conclusions are discussed.

Comment 3 from reviewer 1:

There are also some minor errors,

In line 77, there are no full spelling of the abbreviations of MCP and PIP. The labels of mechanical schematics in Figure 1 is not clear, and the description about the mechanism in line 84-88 is not sufficient. In Figure 7(a), the labels on the horizontal and vertical axes are incorrectly spelled. Figures 7 and 8 are not referenced in the manuscript. In line 347, Figure 3.25 seems misspelled.

Respond 3 to reviewer 1:

All minor corrections have been made.

I wish to thank the reviewers for their useful comments. In the manuscript the comments for reviewer 1 are all highlighted in yellow, those for reviewer 2 are highlighted in turquoise, and those for reviewer 3 are highlighted in Green and for common comments, they are highlighted in grey.

Reviewer 2 Report

This paper is well-written and worthy of publishing in the journal. 
I have only one issue for the authors:

 Please compare the proposed methodology with the other schemes, and demonstrate the advantages and disadvantages to each other.

Author Response

Comment 1 from reviewer 2:

I have only one issue for the authors:

 Please compare the proposed methodology with the other schemes and demonstrate the advantages and disadvantages to each other.

Respond 1 to reviewer 2:   Thank you for giving us the opportunity to improve our work with your valuable comments and criticisms. At the proposed control method subsection (Line 102), the control methodology is compared with other studies and discussed. It has been expanded with the addition of advantages and disadvantages.\

I wish to thank the reviewers for their useful comments. In the manuscript the comments for reviewer 1 are all highlighted in yellow, those for reviewer 2 are highlighted in turquoise, and those for reviewer 3 are highlighted in Green and for common comments, they are highlighted in grey.

Reviewer 3 Report

The manuscript introduces a combined system identification and data-driven control approach for a hand exoskeleton system, termed DDPC. The authors state DDPC's advantages in the possibility of explicitly embedding constraints at the controller's design stage. The evaluation of DDPC should consider feasibility, efficiency, and robustness.

Observations/suggestions:

- In the introduction, the authors mentioned that "System identification, in addition to control algorithms, is crucial for establishing the rehabilitation processes". I would argue that system identification is a necessary step before any control synthesis, independent of the problem.

- there is a rather "hand waving" discarding of model-based control systems in the introduction when referring to MPC. I remind the authors, for instance, that there are a class of control systems, more precisely nonlinear control systems (e.g. variable structure controllers) which can handle both structured and unstructured uncertainties. This enables robust control of systems in the presence of unmodelled dynamics or other types of uncertainties. Please refer to such methods. 

- Please extend the introductory section with a focus on the different types of control applied to exoskeletons for rehab, rather than only considering MPC - which is still a valuable choice in many problems. The data-driver control literature is also broad (Koopman operator formulations, SINDY etc - see Brunton's Data-Driven Science and Engineering book, for instance), please frame better the contribution.

- please separate the materials and methods in subsections when presenting the hardware system and the model (also, please remove auto-correction under the words of Figure 1 left and increase the size of Figure 1 right panel - maybe stack them??)

- I am a bit surprised that you "limit the system and measurements to not be affected by noise" (line 121) - this is a serious flaw, if you design a data-driven control you want to learn from the intrinsic correlation of your historical data and its temporal evolution, as well as the effects of disturbances, have upon your system - especially when is an exoskeleton worn by a person.

- improve the size and quality of Figure 5

- the evaluation in section 3 startes with a reference to another study which used the "output error method" - please add details on how this model looks lie and the internal workings to relate when assessing the performance of DDPC.

- you tackle a tracking problem, why do Figure 5 to 8 have no reference / expected trajectory compared with the controllers? I see only a relative difference and the errors among them (green)??

- figure 13 is pretty hard to interpret, can you please update the visualization

- Is there a reason that the most elaborate analysis in the last part of the section is only using a step signal as a reference and not a sinewave or more complex patterns?

Overall: the manuscript reads more like a laboratory report, or an undergraduate thesis, with little structure and style of a journal manuscript. I hereby request a major revision of the work. I am surprised, that the manuscript only provided, as the authors claim in the conclusion, " experiments are intended to examine how the parameters affecting the suggested control algorithm influence the success of the control. A data-driven predictive control algorithm is optimization-based and certain constraints" and not as promoted in the abstract " feasibility, efficiency, and robustness". I cannot assess any of the " feasibility, efficiency, and robustness" from the analysis or discussion. Please perform experiments to assess "feasibility, efficiency, and robustness" of the proposed method. 

I would kindly ask the authors to address these points, in the next revision.

Other:

- please use an English language proofreading service as the current version contains many ambiguous and, sometimes, incorrect formulations - this only makes reading and following the storyline harder

- line 121: "imaginary case" is maybe formulated better as a simplified case

- ambiguous formulation "The control algorithm, the theoretical details provided in the previous sections, was practiced in an experimental setup." ???

 - line 279: "Table Error! No text of specified style in document.." ???

Author Response

Comment 1 from reviewer 3: In the introduction, the authors mentioned that "System identification, in addition to control algorithms, is crucial for establishing the rehabilitation processes". I would argue that system identification is a necessary step before any control synthesis, independent of the problem.

Respond 1 to reviewer 3: Thank you for giving us the opportunity to improve our work with your valuable comments and criticisms. The sentence was generalized in light of the reviewer's caution, and the word “rehabilitation processes” was replaced with “controller design”. (Line 51)

Comment 2 from reviewer 3:

- there is a rather "hand waving" discarding of model-based control systems in the introduction when referring to MPC. I remind the authors, for instance, that there are a class of control systems, more precisely nonlinear control systems (e.g. variable structure controllers) which can handle both structured and unstructured uncertainties. This enables robust control of systems in the presence of unmodelled dynamics or other types of uncertainties. Please refer to such methods. 

-Please extend the introductory section with a focus on the different types of control applied to exoskeletons for rehab, rather than only considering MPC - which is still a valuable choice in many problems. The data-driver control literature is also broad (Koopman operator formulations, SINDY etc - see Brunton's Data-Driven Science and Engineering book, for instance), please frame better the contribution.

Respond 2 to reviewer 3:  lines 43 through 50 A few of the nonlinear control methods for exoskeletons are explained and cited in references 11 to 15.

Comment 3 from reviewer 3  -  please separate the materials and methods in subsections when presenting the hardware system and the model (also, please remove auto-correction under the words of Figure 1 left and increase the size of Figure 1 right panel - maybe stack them??)

Respond 3 to reviewer 3:  I add new subsections for hardware and control systems to the material and methods section. Correct the figures as well.

Comment 4 from reviewer 3:  I am a bit surprised that you "limit the system and measurements to not be affected by noise" (line 121) - this is a serious flaw, if you design a data-driven control you want to learn from the intrinsic correlation of your historical data and its temporal evolution, as well as the effects of disturbances, have upon your system - especially when is an exoskeleton worn by a person.

Respond 4 to reviewer 3: I appreciate the correction. I think it related to a poor translation. The system can handle noises and uncertainties because it is currently data-driven and relies on system identification. Corrections that were required have been made.

Comment 5 from reviewer 3:

- the evaluation in section 3 startes with a reference to another study which used the "output error method" - please add details on how this model looks lie and the internal workings to relate when assessing the performance of DDPC.

Respond 5 to reviewer 3: 2.3.1 Lines 125 to 140 of the System Identification section now include a new subsection titled Output Error Method for Identification.

Comment 6 from reviewer 3:  you tackle a tracking problem, why do Figure 5 to 8 have no reference / expected trajectory compared with the controllers? I see only a relative difference and the errors among them (green)??

- figure 13 is pretty hard to interpret, can you please update the visualization

- Is there a reason that the most elaborate analysis in the last part of the section is only using a step signal as a reference and not a sinewave or more complex patterns?

Respond 6 to reviewer 3:These graphs (5 to 8)  compare two approaches. I discuss the identification success by using the OEM as a standard and assessing its relationship with SPbM because I have verified the OEbM in earlier studies. This study referenced as [2].  The controller is not analyzing in this seciton. Only the method of subspace identification is analyzed. Figure 13 is updated.

Comment 7 from reviewer 3: Overall: the manuscript reads more like a laboratory report, or an undergraduate thesis, with little structure and style of a journal manuscript. I hereby request a major revision of the work. I am surprised, that the manuscript only provided, as the authors claim in the conclusion, " experiments are intended to examine how the parameters affecting the suggested control algorithm influence the success of the control. A data-driven predictive control algorithm is optimization-based and certain constraints" and not as promoted in the abstract " feasibility, efficiency, and robustness". I cannot assess any of the " feasibility, efficiency, and robustness" from the analysis or discussion. Please perform experiments to assess "feasibility, efficiency, and robustness" of the proposed method. 

I would kindly ask the authors to address these points, in the next revision.

Respond 7 to reviewer 3: The manuscript has been rearranged taking into account the aforementioned statements and two more experiments have been added.

Comment 8 from reviewer 3:

Other:

- please use an English language proofreading service as the current version contains many ambiguous and, sometimes, incorrect formulations - this only makes reading and following the storyline harder

- line 121: "imaginary case" is maybe formulated better as a simplified case

- ambiguous formulation "The control algorithm, the theoretical details provided in the previous sections, was practiced in an experimental setup." ???

 - line 279: "Table Error! No text of specified style in document.." ???

Comment 8 from reviewer 3All corrections have been made.

I wish to thank the reviewers for their useful comments. In the manuscript the comments for reviewer 1 are all highlighted in yellow, those for reviewer 2 are highlighted in turquoise, and those for reviewer 3 are highlighted in Green and for common comments, they are highlighted in grey.

Round 2

Reviewer 3 Report

I still cannot assess any of the promised "feasibility, efficiency, and robustness" of the analysis or discussion. Please provide some paragraphs in the discussion, based on the graphs and table you showed, on how your approach and solution perform in terms of efficiency, and robustness. Most of the other comments and concerns are addressed in the current revision. Still, a minor English language check is needed.

Author Response

Thank you very much for your consideration, and we appreciate your comments that help us improve our article. 

We added some paragraphs about solution performance in terms of efficiency, and robustness at end of the result and discussion chapter (Between lines 495-514). We also added sentences showing the prominent data for this purpose (Line 390 and 412).

The English language of the manuscript was reviewed and changes were made.